# Comparative Genomics and Transcriptomics Analyses Reveal Divergent Plant Biomass-Degrading Strategies in Fungi

**DOI:** 10.3390/jof9080860

**Published:** 2023-08-18

**Authors:** Jiajia Li, Ad Wiebenga, Anna Lipzen, Vivian Ng, Sravanthi Tejomurthula, Yu Zhang, Igor V. Grigoriev, Mao Peng, Ronald P. de Vries

**Affiliations:** 1Fungal Physiology, Westerdijk Fungal Biodiversity Institute & Fungal Molecular Physiology, Utrecht University, Uppsalalaan 8, 3584 CT Utrecht, The Netherlands; j.li@wi.knaw.nl (J.L.); m.peng@wi.knaw.nl (M.P.); 2USA Department of Energy Joint Genome Institute, Lawrence Berkeley National Laboratory, 1 Cyclotron Rd., Berkeley, CA 94720, USA; alipzen@lbl.gov (A.L.); vng@lbl.gov (V.N.); stejomurthula@lbl.gov (S.T.); yzhang13@lbl.gov (Y.Z.); ivgrigoriev@lbl.gov (I.V.G.); 3Plant and Microbial Biology, University of California Berkeley, Berkeley, CA 94720, USA

**Keywords:** CAZymes, comparative genomics, transcriptome analysis, plant polysaccharide degradation

## Abstract

Plant biomass is one of the most abundant renewable carbon sources, which holds great potential for replacing current fossil-based production of fuels and chemicals. In nature, fungi can efficiently degrade plant polysaccharides by secreting a broad range of carbohydrate-active enzymes (CAZymes), such as cellulases, hemicellulases, and pectinases. Due to the crucial role of plant biomass-degrading (PBD) CAZymes in fungal growth and related biotechnology applications, investigation of their genomic diversity and transcriptional dynamics has attracted increasing attention. In this project, we systematically compared the genome content of PBD CAZymes in six taxonomically distant species, *Aspergillus niger*, *Aspergillus nidulans*, *Penicillium subrubescens*, *Trichoderma reesei*, *Phanerochaete chrysosporium*, and *Dichomitus squalens*, as well as their transcriptome profiles during growth on nine monosaccharides. Considerable genomic variation and remarkable transcriptomic diversity of CAZymes were identified, implying the preferred carbon source of these fungi and their different methods of transcription regulation. In addition, the specific carbon utilization ability inferred from genomics and transcriptomics was compared with fungal growth profiles on corresponding sugars, to improve our understanding of the conversion process. This study enhances our understanding of genomic and transcriptomic diversity of fungal plant polysaccharide-degrading enzymes and provides new insights into designing enzyme mixtures and metabolic engineering of fungi for related industrial applications.

## 1. Introduction

Plant biomass is a magnificent renewable carbon source on Earth, which is of major importance for ecology and is an attractive substrate for the production of biofuels and biochemicals. Plant cell walls mainly consist of polysaccharides (e.g., cellulose, hemicellulose, and pectin), lignin, and small amounts of protein. In addition, starch and inulin are commonly found as storage polysaccharides.

In nature, fungi play a central role in the degradation of plant biomass due to their unique ability to secrete a broad range of extracellular enzymes for decomposition of recalcitrant plant polysaccharides into mono- or oligosaccharides that can be metabolized for their growth and reproduction. Based on sequence and structural similarity, carbohydrate-active enzymes are catalogued into five major families in the Carbohydrate-Active enZymes (CAZymes) database (http://www.cazy.org/ (accessed on 24 February 2023)) [1,2], including glycoside hydrolases (GHs), carbohydrate esterases (CEs), polysaccharide lyases (PLs), glycosyltransferases (GTs), and auxiliary activities (AAs). Fungal plant biomass degradation (PBD) has been extensively studied for several decades due to its essential roles in global carbon cycles and increasing potential in advancing the transition from a fossil-based economy into a more sustainable bio-based economy. The plant biomass-degrading enzymes from these fungi have been widely used in many industrial sectors, such as biofuels, biochemicals, paper and pulp, food, feed, detergents, and textiles [3].

Since polysaccharide composition is highly variable between different plant sources and environmental conditions, fungi need a diverse set of enzymes to degrade them (Appendix A). To control their production, fungi have evolved a sophisticated regulatory system to tailor the energy-costly production and export of CAZymes for efficient use of each specific plant substrate [4]. The outcome of this evolutionary adaptation is that different fungal species show enormous diversity in terms of their CAZyme content, enzymatic production, and transcriptional regulation related to plant biomass degradation. The importance of the regulatory layer is highlighted for example by comparative genomics at the genus or section levels of *Aspergillus*, which revealed that fungal growth profiles on polysaccharides do not necessarily correlate with their PBD abilities inferred from genomic CAZyme content [5,6]. An integrative analysis of genomic content, enzyme activity, and proteomics data of eight taxonomically close *Aspergillus* species suggested that fungi with similar genomic potential for PBD could differ dramatically in CAZyme production and overall enzyme activities [7]. Similar results have been reported for *Trichoderma* and *Penicillium* species [8,9,10,11]. In addition, different species employ different sets of transcription factors (TFs) to regulate the utilization of specific polysaccharides. For instance, L-arabinose release from plant biomass and subsequent intracellular metabolism was controlled by AraR in Eurotiomycetes [12], but regulation was governed by a different unrelated TF, ARA1, in Sordariomycetes and Leotiomycetes [13,14]. More strikingly, although a similar set of CAZymes were involved in PBD of Basidiomycete and Ascomycete fungi, very few of the characterized ascomycete regulators have corresponding one-to-one orthologous genes in Basidiomycetes [4,15].

Fungal enzymatic saccharification of plant polysaccharides results in the release of monomeric and oligomeric sugars, and in turn these small sugars or their metabolic conversion products act as inducer or repressor molecules for boosting or suppressing the production of PBD-related CAZymes [16,17,18,19]. For instance, D-galacturonic acid and L-rhamnose, the main monomeric sugar constituents of pectin, have been shown to induce the expression of pectinolytic enzymes in *Aspergillus niger*, *Trichoderma reesei*, and *Neurospora crassa* [20,21,22,23]. In *A. niger*, the expression levels of xylanases are controlled by two major regulators: induction via transcriptional activator XlnR and repression via transcriptional repressor CreA, depending on different xylose concentrations [17]. However, most of previous (post-)genomics studies on fungal PBD-related CAZymes were restricted to model species and were performed in limited growth conditions, or they solely relied on comparative genomics analysis. New studies based on both genomics and post-genomics of taxonomically distant species will enhance our understanding of the molecular mechanisms and biodiversity of fungal PBD, which will provide new insights into the design of enzyme mixtures and guide metabolic engineering of fungi toward improved plant biomass conversion during related industrial applications.

In this study, we systematically compared the genomic repertoires of PBD-related CAZymes in six evolutionarily diverse species, *A. niger*, *Aspergillus nidulans*, *Penicillium subrubescens*, *T. reesei*, *Phanerochaete chrysosporium*, and *Dichomitus squalens*, as well as their transcriptomic response to nine plant-derived monosaccharides. In addition, the specific carbon utilization ability inferred from genomics and transcriptome data was compared to fungal growth profiles across a wide range of carbon sources.

## 2. Materials and Methods

### 2.1. Fungal Strains

This study focuses on six evolutionarily diverse species: three Eurotiomycetes (*A. niger* N402, *A. nidulans* FGSCA4, and *P. subrubescens* CBS132785/FBCC1632), one Sordariomycete (*T. reesei* QM6a), and two Basidiomycetes (*P. chrysosporium* RP-78 and *D. squalens* FBCC312/CBS432.34). The genomes and related annotations of these six fungal species were described in our previous study [24].

### 2.2. Identification of Orthologs of CAZyme Genes Related to Plant Biomass Degradation across Six Fungal Species

CAZyme annotation and related protein sequences of each species were obtained from JGI MycoCosm Portal (https://mycocosm.jgi.doe.gov (accessed on 24 February 2023)). The PBD-related CAZy families (based on previous studies) were selected for further analysis [7,25,26]. Orthologous genes were identified using OrthoMCL [27] (https://orthomcl.org/orthomcl/app (accessed on April 14 2023)) and OrthoFinder [28] (https://github.com/davidemms/OrthoFinder (accessed on April 14 2023)) with the protein sequences as input and default parameters applied. We took the combination of the two ortholog mapping results as our final ortholog set.

### 2.3. Phylogenetic Analyses

Protein sequences of each selected CAZy family (PBD-related) were aligned using MAFFT v7.310 [29,30]. Unusually long and incomplete sequences were corrected manually based on NCBI BlastX searching results, whereas duplicate and ambiguous sequences were discarded. Phylogenetic trees generated using FastTree [23] were used to distinguish true orthologs across six fungi. The resulting trees were visualized using iTOL [31].

### 2.4. Transcriptome Sequencing and Analysis

The transcriptome data of *D. squalens* was obtained from our previous study (Gene Expression Omnibus (GEO) database accession: GSE105076) [18]. Transcriptome data of *A. niger* was newly generated in this study and was deposited in the Sequence Read Archive at NCBI under the accession numbers: SRP448993, SRP449003–SRP449007, SRP449023, SRP449039, SRP449049, SRP449062, SRP449079–SRP449081, SRP449083–SRP449085, SRP449089, SRP449068–SRP449070, SRP449098, SRP449125, SRP449138, SRP449141, SRP449142, SRP449151, and SRP449193, while the other four species grown on nine monosaccharides were recently generated in our previous study [24].

The culture conditions were maintained as similarly as possible to allow a good comparison of the species. In detail, *A. niger* NRRL3, *A. nidulans* FGSC A4, and *P. subrubescens* FBCC1632/CBS132785 were pre-cultured in complete medium [32] with 2% D-fructose, and mycelial aliquots were then transferred to minimal medium [32] with 25 mM D-glucose, D-fructose, D-galactose, D-mannose, L-rhamnose, D-xylose, L-arabinose, D-galacturonic acid, and D-glucuronic acid, respectively, and cultivated for 2 h. The same cultivation approach was used for *T. reesei* QM6a and *P. chrysosporium* PR-78, but with media optimized for these species [33,34] and cultivated for 4 h. The cultivation approach for *D. squalens* can be found in our previous study [18]. So, only the basic salt medium is specific for the species, but the carbon source is the same in all species. Mycelial samples were harvested and immediately frozen in liquid nitrogen.

Total RNA was extracted from ground mycelial samples using TRIzol reagent (Invitrogen). Purification of mRNA, synthesis of cDNA library, and sequencing were conducted at DOE Joint Genome Institute (JGI). The details of transcriptome data generation and related downstream analysis were described previously [24].

In addition, the CAZy genes with strong sugar specificity were selected using a sugar specificity index (SSI), which was calculated by following the equation that has been widely used in studies of gene expression specificity [35,36]:$$SSI=\frac{\sum_{i=1}^{n}\left(1-\hat{x_{i}}\right)}{n-1};\hat{x_{i}}=\frac{x_{i}}{\underset{1\leq i\leq n}{\max}\left(x_{i}\right)}$$

For the above equation, *x_i_* is the expression of the gene in each specific monosaccharide condition *i*, and *n* is the number of total tested monosaccharide conditions. We define the CAZyme genes with *SSI* > 0.7 as sugar-specific genes (SSGs). In addition, we filtered out the extremely low-expressed genes whose maximum expression levels (FPKM) were less than 10 in all tested conditions. For each SSG, we defined the corresponding sugar growth condition in which the gene showed highest expression level among all tested sugars as its inducing sugar.

### 2.5. Growth Profiling on 18 Plant Biomass-Related Substrates

For growth phenotype analyses, all strains were grown on minimal medium (MM) [32] with monosaccharides, disaccharides at 25 mM, and polysaccharides at 1% final concentration. Growth was performed at 30 °C for two *Aspergillus* spp. and 25 °C for the other species. Media with no carbon source was used as a control. If growth on a specific carbon source is the same as on no carbon source, it is considered as no growth. Growth on D-glucose was used as an internal reference for growth. Substrates were obtained from Sigma-Aldrich (Zwijndrecht, The Netherlands): D-glucose (G-8270), D-fructose (F-0127), D-galactose (G-0625), D-mannose (M-2069), L-rhamnose (83650), D-galacturonic acid (48280), D-glucuronic acid (G-8645), sucrose (S-5016), maltose (M-9171), beechwood xylan (X-4252), guar gum (G-4129), apple pectin (76282), inulin (I-2255), and cellulose (C8002); from Difco (Leeuwarden, The Netherlands): soluble starch (217820); or from Acros (Geel, Belgium): cellobiose (108465000) and citrus pectin (416862500).

## 3. Results

### 3.1. Genomic Potential of PBD CAZymes Shows Diversity across Six Fungi

In total, the genome repertoire of 65 CAZy (sub-)families and feruloyl esterases (FAEs) involved in degradation of plant polysaccharides was comparatively analyzed in six fungi (Figure 1 and Appendix A). Specifically, the selected PBD-related enzymes studied included four auxiliary activities (AAs) families (AA3, AA9, AA13, and AA16), six carbohydrate esterase families (CE1, CE5, CE8, CE12, CE15, and CE16), five polysaccharide lyase families (PL1, PL3, PL4, PL9, and PL11), 39 glycoside hydrolase (GH) families, and FAEs (Figure 1). Notably, ten enzyme families (GH3, GH12, CE16, GH2, GH35, GH31, GH43, GH51, GH54, and FAE) are multifunctional and involved in the degradation of more than one plant polysaccharide (Appendix A).

The potential for PBD as inferred from the genomic repertoire of PBD CAZymes shows clear diversity across the studied fungi (Figure 1 and Appendix A). Overall, the CAZyme repertoire differences correlate well with the taxonomic distance of the studied fungi. The three Eurotiomycetes (*A. niger*, *A. nidulans*, and *P. subrubescens*) possess a relatively high number of PBD-related CAZymes compared to *T. reesei* and the Basidiomycetes (*P. chrysosporium* and *D. squalens*), especially with respect to enzymes from GH and PL families (Appendix A and Appendix A). *T. reesei* contains fewer CAZymes involved in each plant polysaccharide than the other three Ascomycetes. The two Basidiomycetes have fewer hemicellulolytic, pectinolytic, and starch-degrading enzymes, while the total numbers of cellulolytic enzymes are comparable to Eurotiomycetes. Notably, a higher number of AA9 genes were identified in the Basidiomycetes. In the following sections, we discuss in more detail the diversity of genomic potential with respect to degradation of each polysaccharide.

#### 3.1.1. Cellulose Degradation

Cellulose is a major component of plant biomass. The degradation of cellulose mainly involves four groups of enzymes: endoglucanases (EGLs), cellobiohydrolases (CBHs), and β-glucosidases (BGLs), as well as auxiliary activities enzymes (AAs) (Appendix A) [37].

EGLs act mostly on amorphous regions of cellulose, releasing reducing and nonreducing chain ends [38,39]. They are mainly classified in GH12, GH131, and GH45 and three subfamilies of GH5 (GH5_4, GH5_5, and GH5_22) (Appendix A). Many of these EGL families showed noticeable differences between both evolutionarily close and distant species, especially the GH5 (sub-)family. GH5_4 has a single copy in *A. nidulans* and *P. subrubescens* but is missing in all other species. GH5_22 was not identified in the two *Aspergillus* species and *T. reesei* but is present as two copies in *P. subrubescens* and the two Basidiomycetes. *P. subrubescens* has more copies of GH5_5 and GH12 than the other species. In contrast to most EGLs with comparable or more copies in Ascomycetes than Basidiomycetes, GH45 and GH131 have multiple copies in both Basidiomycetes, but are present as a single copy in three Ascomycetes (except for the absence of GH131 in *T. reesei*).

CBHs release cellobiose from the reducing (CBH I) or non-reducing (CBH II) ends of the cellulose. Fungal CBH I and CBH II genes are classified in GH7 and GH6, respectively. Gene copies of GH7 are notably larger (eight) in *P. chrysosporium* and vary among the other species (two to four). GH6 was present as a single copy in *T. reesei* and the Basidiomycetes and with two genes in the three Eurotiomycetes.

BGLs are involved in the cleavage of cellobiose into glucose and are mainly grouped into the GH1 and GH3 families. GH3 BGLs are greatly expanded in the three Eurotiomycetes, while GH1 is only significantly expanded in *P. subrubescens*.

LPMOs and cellobiose dehydrogenases (CDHs) have been suggested to be involved in the oxidative cleavage of cellulose and boosted cellulose degradation [26,40,41]. AA9 is a widely distributed LPMO in fungi, which was present as multiple copies in all studied fungi and was significantly expanded in the Basidiomycetes compared to the Ascomycetes. In contrast, AA16 LPMO was only present as a single copy in the Eurotiomycetes and was missing in the other species, while AA3_1 CDH showed less variation among the studied fungi.

#### 3.1.2. Hemicellulose Degradation

In contrast to cellulose, hemicellulose is a heterogeneous polysaccharide, which is composed of different residues that form diverse backbones and branches. Degradation of plant hemicelluloses requires a variety of hemicellulolytic enzymes, which are divided over at least twenty GH families, four CE families, and FAE (Figure 1 and Appendix A and Appendix A).

Xylan degradation

Xylan is a common hemicellulose polysaccharide. To degrade xylans, a specific set of CAZymes is required, mainly including: β-D-endoxylanase (XLN) from GH10 and GH11 [42]; β-xylobiohydrolase (XBH) from GH30_7; BXL from GH3 and GH43; α-L-arabinofuranosidase (ABF) from GH43, GH51, and GH54; α-glucuronidase (AGU) from GH67 and GH115; glucuronoyl esterase (GE) in CE15; acetyl xylan esterase (AXE) from CE1 and CE5; hemicellulose acetyl esterase (HAE) from CE16 [43]; arabinoxylan arabinofuranohydrolases (AXH) from GH62; and feruloyl esterase (FAE). Overall, xylanolytic enzyme-encoding genes in the six studied species differ significantly. The genome of *P. subrubescens* contains the largest number of xylanases, followed by the two *Aspergillus* species, while *T. reesei* and the two Basidiomycetes had the lowest number. Higher gene copies in GH3, GH43, GH51, and GH54 families were found in the Ascomycetes than the Basidiomycetes, while GH10 and CE16 showed opposite trends. In addition, CAZy families GH54 (ABF), GH62 (AXH), GH67 (AGU), and CE5 (AXE) were exclusively found in the Ascomycetes. CE15 is absent in the Eurotiomycetes but present in the two Basidiomycetes and *T. reesei*, while more copies of FAEs and GH43 were identified in Eurotiomycetes than in the other species.

Xyloglucan degradation

Xyloglucan has complex structure with various side chains and decorations [44]; thus, its hydrolysis requires a vast arsenal of enzymes. Except for ABF and HAE enzymes mentioned earlier, xyloglucan-degrading enzymes also include α-fucosidase (AFC) from GH29, GH95 [45], and GH141; xyloglucan β-1,4-endoglucanase (XEG) from GH12 [46], GH44 [47], and GH74 [48]; α-xylosidase (AXL) from GH31 [45]; and β-1,4-galactosidase (LAC) from GH2 and GH35 [37]. High variety was observed in the copy number of these CAZy families. *P. subrubescens* has expanded several xyloglucan-degrading enzyme families, such as GH12, GH43, and GH54. Numbers in families GH95, GH12, GH35, GH2, and GH31 varied remarkably in the Ascomycota but showed low variation between the two Basidiomycetes (Figure 1). In contrast, GH29 was exclusively identified in *A. niger* and *P. subrubescens,* while GH44 and GH141 were only identified in *D. squalens* and *P. subrubescens*, respectively.

Mannan degradation

The degradation of mannans requires the concerted action of several mannanases, including β-1,4-endomannanase (MAN), β-1,4-mannosidase (MND), α-galactosidases (AGLs), and HAE. Fungal AGLs have been assigned to GH27 and GH36. MANs are mainly classified in GH5_7 with fewer copies in GH26 and GH134. MND and HAE belong to GH2 [49] and CE16, respectively (Figure 1).

Overall, the genomes of *A. nidulans* and *P. subrubescens* contain more copies of mannanases than the other fungi (Figure 1). The differences are that *A. nidulans* has more copies of MAN (GH26, GH134, and GH5_7), whereas *P. subrubescens* harbors large numbers of AGL (GH27 and GH36)- and MND (GH2)-encoding genes. In general, the other four species have fewer mannanases, except that *T. reesei* and *A. niger* contain considerable numbers of AGLs (especially GH27) and MNDs, and the Basidiomycetes have a higher number of CE16 copies. Notably, GH36, GH26, and GH134 were absent in both Basidiomycetes (Figure 1), and GH26 and GH134 were absent in the Basidiomycetes and *T. reesei*.

#### 3.1.3. Pectin Degradation

Pectin is a very complex polymer, composed of a family of galacturonic acid-rich polysaccharides, including: homogalacturonan (HG), xylogalacturonan (XGA), and two types of rhamnogalacturonan (RG-I and RG-II) [50]. Its degradation requires a variety of enzymes (Appendix A), which are classified into more than 20 CAZy families, mainly including pectin hydrolases (GH2, GH5_16, GH28, GH35, GH43, GH51, GH54, GH78, GH88, GH93, and GH105), pectin lyases (PL1, PL3, PL4, PL9, and PL11) [51], pectin methylesterases (CE8), and pectin/rhamnogalacturonan acetyl esterase (CE12) [52]. The gene content of pectinolytic genes showed huge differences among the six studied species. In general, the Eurotiomycete fungi have a higher number and more diverse set of pectinolytic genes than the other three studied species (Figure 1). Several enzyme families involved in pectin degradation are reduced or absent in *T. reesei*, including the absence of four GH families (GH51, GH53, GH88, and GH93) and many fewer genes encoding for GH28 and GH78 compared to the other three Ascomycetes. PLs are exclusively present and largely varied in three Eurotiomycota species, of which *A. nidulans* has more types of PL families (e.g., PL1, PL3, PL4, PL9, and PL11). The other studied species lost most of the PLs, except *D. squalens*, which has one gene of PL4_3. Regarding pectin methylesterases, CE8 was found in multiple copies in most species except in *T. reesei*, while CE12 was present in the three Eurotiomycota species and *D. squalens* and was absent in *T. reesei* and *P. chrysosporium*.

#### 3.1.4. Storage Polysaccharides Degradation

Starch and inulin are two main storage polysaccharides of plants. Starch consists of an α-1,4-linked polymer (amylose) of D-glucose residues that can be branched at α-1,6-linked points (amylopectin) [53,54,55]. It is degraded by the joint enzymatic actions of α-glucosidase (AGD) from GH13_40 and GH31, amylo-α-1,6-glucosidase (AMG) from GH133, α-amylase (AMY) from GH13_1 and GH13_5, glucoamylase (GLA) from GH15, and LPMOs from AA13 (Appendix A). Most starch-degrading CAZymes were identified with one or more copies in all studied species, except that AA13 was only identified in *A. nidulans* and *P. subrubescens,* and GH13_5 and GH13_40 were absent in *T. reesei* and *D. squalens*, respectively. Additionally, the three Eurotiomycetes contain more copies of GH13_1 than the other species, while *P. subrubescens* harbors the largest number of GH13_40 and GH15 (Figure 1).

Inulin consists of a branched β-2,1-linked chain of D-fructose with a terminal D-glucose residue [56,57]. GH32 is the only CAZy family involved in inulin degradation, and it mainly acts as endo-inulinase, exo-inulinase, and invertase. GH32 enzymes were absent in *T. reesei* and *P. chrysosporium* but were expanded in *A. niger* and *P. subrubescens* with five and nine copies, respectively (Figure 1).

### 3.2. Expression Profile of CAZy Genes during Fungi Grown on Different Monosaccharides

PBD CAZymes mediate the release of monomeric/oligo-sugars from plant polysaccharides. In turn, the presence of these small sugars can induce or repress the expression of CAZyme-encoding genes. Here, we investigated the transcriptome diversity of CAZymes in response to different plant-derived monosaccharides of studied fungi.

Overall, the expression profiles of CAZy genes varied across the tested species and enzyme families. The inducing pattern of pectinolytic genes showed considerable consistency across the studied species (Figure 2). For most of the studied Ascomycetes, relative high abundance of pectinolytic genes were observed on L-arabinose, D-galacturonic acid, D-glucuronic acid, or L-rhamnose, which are the most abundant sugars present in the polymers of pectin. *A. nidulans* is an exception, in which pectinolytic genes mainly had higher expression on L-rhamnose and were not clearly induced by D-galacturonic acid (Figure 2 and Appendix A). Additionally, the genes involved in the degradation of xylan and xyloglucan showed relatively high expression on L-arabinose in the Ascomycetes, particularly *P. subrubescens*, which had more highly expressed genes on L-arabinose, while they were also induced by D-xylose in *A. niger* and induced by L-rhamnose in *A. nidulans*. Notably, no clear sugar-inducing pattern was observed for the studied Basidiomycetes, except that most PBD-related CAZymes showed relatively high expression levels on D-galacturonic acid compared to the other tested sugars for *P. chrysosporium*.

In addition, we also analyzed the sugar specificity index of each CAZy gene. Sugar specificity is often described based on gene expression levels, and we define genes with a high sugar specificity index as a group of genes whose function and expression are specific for one certain sugar condition. This index can be used to evaluate if the specific CAZy genes involved in certain polysaccharide degradation showed corresponding inducing patterns on the corresponding monosaccharides (Appendix A and Appendix A). As expected, in the three Eurotiomycetes, the CAZy genes involved in pectin degradation showed clear D-galacturonic acid- and L-rhamnose-inducing specificity (except mainly L-rhamnose-inducing patterns in *A. nidulans*), and the CAZy genes involved in xylan or xyloglucan degradation revealed L-arabinose- and D-xylose-inducing patterns. Notably, parts of CAZy genes involved in the degradation of the above polysaccharides also showed high expression levels on D-galactose in *A. niger* and *A. nidulans*, but not in *P. subrubescens*, which indicates metabolic cross talk between sugar-sensing pathways and/or overlap in regulatory networks. In addition, ten and eighteen CAZymes involved in pectin, xylan, or mannan degradation showed sugar-specific induction on D-glucuronic acid in *A. nidulans* and *P. subrubescens*, respectively. In contrast, *T. reesei* only had a small number of CAZy genes with a high sugar specificity index, such as six CAZy genes involved in xylan and xyloglucan that showed L-arabinose induction and three pectinolytic genes with D-galacturonic acid induction specificity (Appendix A). We detected more than twenty CAZymes highly expressed on D-galacturonic acid in *P. chrysosporium*, but only nine genes belong to pectinolytic genes. In *D. squalens*, due to the limited transcriptome data, we only observed a very small number of genes showing sugar specificity (Appendix A and Appendix A).

### 3.3. Growth Profiles Do Not Always Correlate with the (Post-)Genomics Profiles of CAZy Genes

The noticeable diversity of genomics and expressional profiles of CAZy genes identified above in the tested fungi strongly suggests that they employ different PBD strategies. To evaluate the correlation between (post-)genomic profiles and actual carbon utilization ability, we analyzed the growth profile of these fungi on nine monosaccharides, two disaccharides, and seven polysaccharides (Figure 3A). Growth on D-glucose and no carbon source were used as an internal reference for growth. The relative difference between growth on different carbon sources was examined between the species.

In general, the Eurotiomycetes showed better growth on most of the tested polysaccharides than the other species, which is consistent with a higher gene content of the corresponding degrading genes (Figure 3B). In contrast, possibly due to the absence and reduction of many CAZy genes in the genome, *T. reesei* grew relatively poorly on most of the tested polysaccharides (except starch). Different species showed huge variation of growth on inulin. *A. niger* and *P. subrubescens* grew well on inulin, whereas the other four species grew very poorly. This correlates well with the diversity of inulin-degrading genes in different species. *A. niger* and *P. subrubescens* have many copies of inulinolytic genes, in contrast to the other species that are impaired in their inulin-degrading machinery. Compared to the other two analyzed Eurotiomycetes, *A. nidulans* had fewer copies of GH32, and it was particularly missing copies of endo-inulinase (Appendix A), which reduces its ability to efficiently utilize this substrate as a carbon source. In *T. reesei* and *P. chrysosporium*, GH32 was fully absent. In *D. squalens*, two copies of GH32 were identified, but their expression was extremely low (Appendix A).

Besides these consistent observations for genomic repertoire, expression, and growth, inconsistencies were also observed. Good growth on citrus pectin was observed for *T. reesei* and *P. chrysosporium*, while their genomes encode fewer pectinolytic genes than the Eurotiomycetes. Additionally, we observed active gene expression of pectinolytic genes for these two species on D-galacturonic acid, the major sugar component of pectin (Figure 2 and Figure 3C). In contrast, *D. squalens* grew poorly on pectin. In line with the crucial role of pectin lyases (PLs) on degrading apple pectin [58], the absence of PLs on the genome of *T. reesei* and *P. chrysosporium* could explain their relatively poor growth on apple pectin. All tested fungi could grow relatively well on starch, while none of the studied species grew well on cellulose as the sole carbon source despite a significant number of cellulases encoded in their genomes. The Eurotiomycetes grew well on xylan, while the other three tested species showed only minimal growth on xylan, which is consistent with the fact that fewer xylanase genes were identified in their genomes and actively expressed during growth of these fungi on xylan-related monosaccharides (Figure 3). In contrast, we observed poor correlation between (galacto-)mannan (guar gum) and mannan-degrading CAZyme content in the fungal genomes. The Eurotiomycetes grew well on guar gum, even though *A. niger* has a very small number of mannan-degrading genes. Compared to *A. niger*, *T. reesei* and the Basidiomycetes have a comparable number of mannan-degrading genes, but they all showed impaired growth. Unexpectedly, none of the fungi showed expression specificity on mannose (Figure 3C and Appendix A), but some showed high sugar specificity on other monosaccharides, like *A. niger* and *P. subrubescens* on L-arabinose. For *A. niger*, it has been shown that mannobiose, rather than mannose, is the inducer of the mannanolytic enzyme system [59], which could explain why there is no mannose-specific induction observed.

## 4. Discussion

The genome content of CAZymes showed considerable differences among the tested species, especially among the taxonomically distant species. The CAZymes’ repertoire and their expression under different carbon sources of each specific fungus imply the possible different PBD approaches. In general, the studied Eurotiomycetes possess more diverse sets of PBD genes that enable them to grow well on most tested polysaccharides, while *T. reesei* and the two Basidiomycetes grew poorly on most polysaccharides due to a relatively small repository or less actively expressed PBD genes. Particularly, *T. reesei* is the only tested Ascomycete that showed impaired growth on maltose and starch, which correlates with fewer starch-degrading enzymes identified in the genome of this species.

Among the three tested Eurotiomycota, only *A. niger* and *P. subrubescens* can grow well on both sucrose and inulin, while *A. nidulans* showed impaired growth on inulin. The divergence could be explained by the lower number of copies of GH32, and the low expression level of intracellular invertase SucB (Aspnid1|2301) [60].

The detailed structures and compositions of polysaccharides significantly affect their degradation by specific fungi, especially for species with less variety of enzyme families encoded in their genomes, which was demonstrated by the growth profile of *T. reesei* and *P. chrysosporium* on pectin extracted from two different plant sources.

Although crucial PBD CAZy genes were present in their genomes, the two tested wood-degrading Basidiomycetes showed relatively poor growth on most tested polysaccharides and showed clear different gene expression profiles of PBD genes than the tested Ascomycetes. The possible reason could be that these fungi mainly grow on wood in nature and have adapted their induction system to grow on complex polysaccharides as carbon sources instead of being optimized for individual polysaccharides as we tested here. In addition, other observations have suggested that inducers of CAZy genes in Basidiomycetes may be disaccharides, rather than the monosaccharides tested here (M. Mäkelä & R.P. de Vries, unpublished results).

Poor growth on cellulose was observed for all tested species, although cellulolytic genes were identified in their genomes. One possible explanation is that the necessary enzymes were not fully activated during fungal growth on cellulose. For instance, many cellulolytic genes in *A. niger* are controlled by the xylanolytic transcriptional activator XlnR, which was mainly activated in the presence of xylose [61,62]. In line with this, visible growths on the cellulose-derived oligosaccharide, cellobiose, were observed for all six species (Figure 3A), which require many fewer degrading enzymes to fully decompose than intact cellulose. In addition, commercial cellulose is not structurally identical to natural cellulose, due to the method of extraction.

We identified several CAZy families that are uniquely present in certain species or phyla, which could indicate a special physiological role and potential industrial applications. For example, the AA16 family only exists in the three Eurotiomycetes, while the AA9 family is broadly present in all six species. A recent study suggested these two fungal LPMO families are counterparts and target the same substrates [26]. Contradicting this, a recent study found that AA16 can only boost the cellulase activity of AA9 but cannot break down cellulose independently [63]. Clearly, more detailed studies in this area are warranted. In addition, GH44 enzymes were only identified in *D. squalens* [47], which indicates the scarcity and unique function of this enzyme. Additionally, we found CE15 only in the two Basidiomycetes and *T. reesei*, which was proposed to cleave ester linkages between lignin and glucuronoxylan [64]. GH141, whose active site pocket of the hydrolase is likely to be significantly more open than other fucosidases (e.g., GH95 and GH29), was only present in *P. subrubescens*, revealing that it has a more variable PBD mode [65]. Only *A. nidulans* contains genes encoding PL3 and PL9 pectate lyases, as well as PL11 rhamnogalacturonan lyases, which indicates that *A. nidulans* may have great potential for commercial applications in pectin degradation [66].

In addition, some genes encoding a specific family present phyla specificity. GH54 is only present in the four Ascomycetes and has been reported to remove decorations including L-arabinose and D-galactose in the presence of metal cofactors, which may suggest that it could be introduced in enzymatic cocktails to assist in the hydrolysis of xylan and xyloglucan. Similarly, GH62 was only found in the four Ascomycetes, and it is the sole GH family that contains only arabinofuranosidase (ABF) specificities [67]. However, high variability was also observed in the number of genes within the same phyla. For example, multiple gene copies of GH51-, GH54-, and GH62-encoding ABFs were identified in *P. subrubescens,* which implies a potential enhancement in the ability of *P. subrubescens* to release L-arabinose from plant biomass in a more specific manner [68]. Moreover, this is in line with our finding that CAZy genes involved in xylan, pectin, and xyloglucan in *P. subrubescens* show evident L-arabinose specificity (Figure 3C).

Additionally, the expression profiles of PBD genes in six tested species showed dramatic differences during fungal growth on nine monosaccharides (Figure 2 and Figure 3C). For the closely related Eurotiomycetes, although they have comparable content of pectinolytic genes, the varying expression patterns of these genes suggest diversity in the underlying complex regulatory networks. In *A. niger,* a comparable set of pectinolytic genes were induced on L-arabinose, L-rhamnose, and D-galacturonic acid, while the same set of genes in *A. nidulans* were largely induced by L-rhamnose alone, and in *P. subrubescens* they were mainly induced by L-arabinose and D-galacturonic acid (Appendix A). Similarly, sugar-specific CAZymes involved in xylan or xyloglucan degradation in different fungi also displayed different inducing patterns on L-arabinose and D-xylose.

## 5. Conclusions

In this study, we revealed a high diversity of fungal plant polysaccharide degrading enzymes at both the genomic and transcriptomic levels. Considering the importance of the ecological role and increasing industrial potential of fungal plant biomass degradation, the findings revealed in this study will improve our understanding of fungal diversity and provide new insights into designing enzyme mixtures and metabolic engineering of fungi for related industrial applications.

## Figures and Tables

**Figure 1 jof-09-00860-f001:**
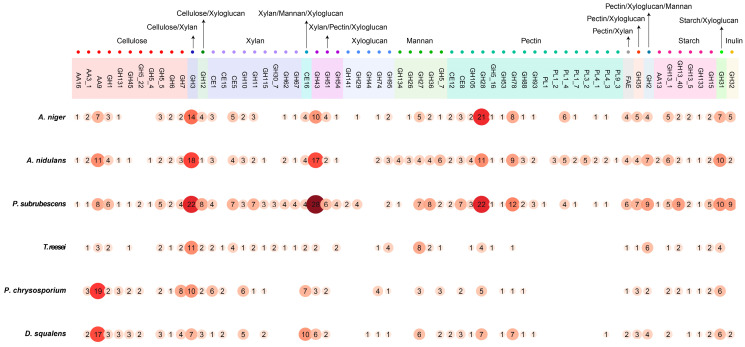
Distribution of gene numbers in selected CAZy families in the six fungi. Bubbles with numbers represent the number of encoding genes in each CAZy family. Color intensity and bubble size are related to the numbers within them.

**Figure 2 jof-09-00860-f002:**
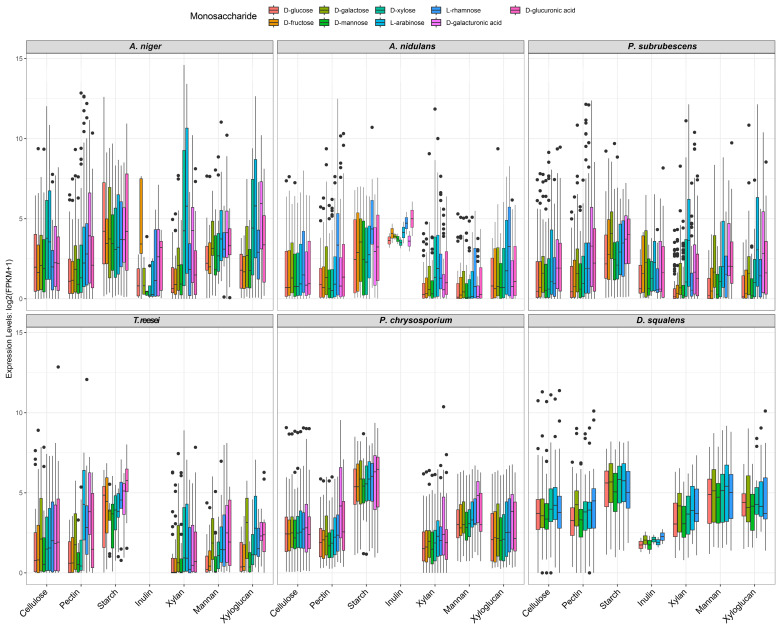
Distribution of the expression profiling of the CAZy genes involved in degradation of different polysaccharides (mentioned at the bottom) in the tested fungi. On the top, different colors indicate different monosaccharides that were used as carbon sources in the fungal cultures. The black dots indicate outlier genes with extremely low or high expression values.

**Figure 3 jof-09-00860-f003:**
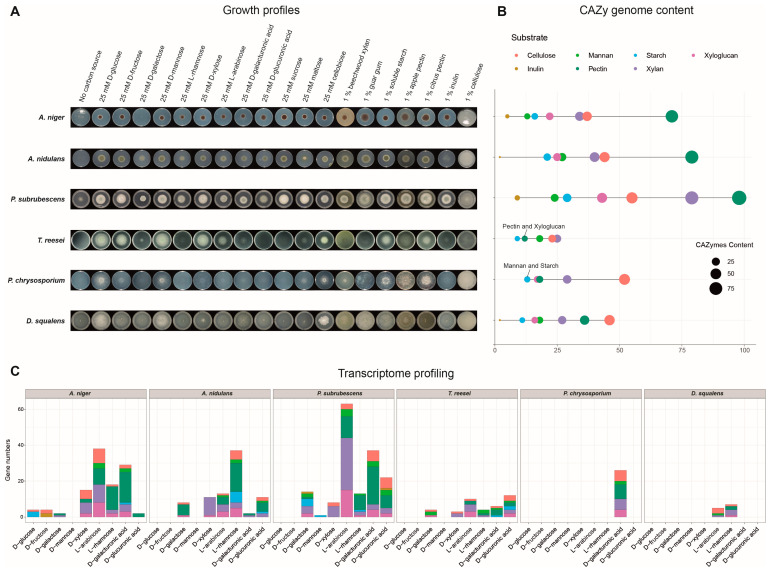
Overview of growth profiles, CAZyme content, and transcriptome profiling in the studied fungi. (**A**) Growth profiles on various plant biomass-related carbon sources. (**B**) Summary of CAZy genome content involved in the degradation of each plant polysaccharide. The size of the dots indicates the number of CAZy genes, and the color indicates different substrates that CAZymes act on. (**C**) Distribution of CAZy genes with sugar-specific expression on each monosaccharide. The same color code as Figure 3B was used to indicate the different substrates.

## Data Availability

The reads from each of the transcriptome sequencing (RNA-seq) samples were deposited in the Sequence Read Archive at NCBI under the accession numbers: *A. niger* SRP448993, SRP449003–SRP449007, SRP449023, SRP449039, SRP449049, SRP449062, SRP449079–SRP449081, SRP449083–SRP449085, SRP449089, SRP449068–SRP449070, SRP449098, SRP449125, SRP449138, SRP449141, SRP449142, SRP449151, and SRP449193; *A. nidulans* SRP262827-SRP262853; *P. subrubescens* SRP246823-SRP246849; *T. reesei* SRP378720-SRP378745; and *P. chrysosporium* SRP249214-SRP249240.

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
