# Peer review of "Comparative Genomics and Transcriptomics Analyses Reveal Divergent Plant Biomass-Degrading Strategies in Fungi"

_jof, 2023, doi:10.3390/jof9080860_

Round 1

Reviewer 1 Report

The manuscript presents a comparative study of six species of fungi in terms of their content of plant biomass-degrading (PDB) carbohydrate- active enzymes (CAZymes) in their genome. They also performed a transcriptome analysis of the six species grown on different carbon sources (in this case, nine monosaccharides in order to compare the diversity of expressed CAZymes.  The work has high relevance in modern world since fungi-derived CAZymes are important tools for plant biomass degradation to produce biofuels and other products in the context of biorefineries. The authors chose This study focusses on evolutionarily diverse species, Eurotiomycetes, one Sordariomycete (T. reesei) and Basidiomycetes. The genome and annotation of those species  had been carried out previously by the same group.

Overall, the manuscript was very well written without any language mistake that I could notice. The figures and supplementary material help the understanding of the results that confirm the diversity the diversity of PDB CAZymes on genomic and transcriptomic level.  

Therefore, in my opinion the manuscript should be accepted in the present form.

Author Response

We thank the reviewer for her/his positive comments

Reviewer 2 Report

Manuscript entitled “Comparative genomics and transcriptomics analyses reveal divergent plant biomass degrading strategies in fungi” by Jiajia Li et al. is a well written article improving our understanding of the fungal diversity and providing new insights into enzyme designing.  Manuscript flows well from start to end. I recommend consideration for publication it in journal Diversity.

Nevertheless, there are some minor revisions that I would like to point out.

1.      It is necessary to indicate the full biological name of fungi including authors at their first mentioning

2.      Also the name of the strains used in the work should be written in the chapter Materials and Methods.

3.      In the materials and methods the authors are not sure about the uniformity of media for all fungi used in the analysis. And still the media used for the analysis were the same or not? This is very important considering that the studied enzymes are induced by substrate.

4.      Figure 3a is hard to read. The results of incubating the fungi with minimal growth zone are not visible

Str. 18 change Taxonomic to Taxonomically

33 Earth from the capital letter

51 change “across” to “between”

129 change to ”D-galacturonic”

152 change “Aspergilli” to “Aspergillus spp.”

Author Response

  1. It is necessary to indicate the full biological name of fungi including authors at their first mentioning

> This has been checked and corrected

  1. Also the name of the strains used in the work should be written in the chapter Materials and Methods.

> This has been added.

  1. In the materials and methods the authors are not sure about the uniformity of media for all fungi used in the analysis. And still the media used for the analysis were the same or not? This is very important considering that the studied enzymes are induced by substrate.

> The only difference in the conditions between the strains is the basic salt medium that was used. The media for the different species has been described previously and these references are already present in the text. These media do not contain another carbon source than the one we added, and therefore we can consider them similarly as they only support the basic mineral requirements of the strains and do not affect the carbon source related induction. A sentence has been added to specify this.

  1. Figure 3a is hard to read. The results of incubating the fungi with minimal growth zone are not visible

> Figure 3a only serves to show the difference in growth of the fungi on the substrates. While indeed differences between plates with poor growth are hard to see, this is not really the main importance of the figure. The main point of the figure is to compare good to poor growth and these differences are clearly visible (e.g. for P. chrysosporium on glucose, inulin and citrus pectin). It would be an option to make this panel into a separate figure and make it larger, but we don’t feel that is really essential for understanding its relevance to the study.

Str. 18 change Taxonomic to Taxonomically

> This has been changed

33 Earth from the capital letter

> This has been changed

51 change “across” to “between”

> This has been changed

129 change to ”D-galacturonic”

> This has been changed

152 change “Aspergilli” to “Aspergillus spp.”

> This has been changed